# Beyond Biodiversity: Eliciting Diverse Values of Urban Green Spaces in Flanders

Thomas Bastiaensen [1,*], Ewaut Van Wambeke [2], Camelia El Bakkali [2,3], Jomme Desair [2], Charlotte Noël [2], Kaat Kenis [4], Lukas Vincke [5] and Sander Jacobs [1,2]

1    Terrestrial Ecology Unit, Ghent University, 9000 Ghent, Belgium; sander.jacobs@inbo.be
2    Research Group Nature and Society, Research Institute for Nature and Forest INBO, Havenlaan 88 bus 73, 1000 Brussels, Belgium; ewaut.vanwambeke@inbo.be (E.V.W.); camelia.el.bakkali@vub.be (C.E.B.); jomme.desair@inbo.be (J.D.); charlotte.noel@inbo.be (C.N.)
3    Community Ecology Laboratory, Vrije Universiteit Brussel, Pleinlaan 2, 1050 Brussels, Belgium
4    Department of Product Development, University of Antwerp, 2000 Antwerp, Belgium; kaat.kenis@student.uantwerpen.be
5    Department of Biology, University of Antwerp, 2000 Antwerp, Belgium; lukas.vincke@student.uantwerpen.be
*    Correspondence: thomas.bastiaensen@ugent.be

**Abstract:** Nature-based solutions are claimed to offer an effective approach to tackle societal challenges and promote biodiversity. While research has mainly focused on biodiversity and material ecosystem services, non-material contributions and relational values of urban green spaces remain underexplored. How to balance the benefits of nature, well-being, and relational values in their design and performance evaluation remains unclear. To elicit the values expressed in public communication regarding the benefits of urban nature projects in Flanders, three online repositories that feature diverse nature-based solutions projects in the region were chosen. Using coding and quantitative content analysis of standardized descriptions from these repositories, this study found that relational values were most abundant (55%), followed by instrumental values (30%) and intrinsic values (15%), consistently so over socio-demographic and physical contexts. It was also discovered that larger projects have a higher level of multifunctionality, which is calculated based on the variety of values and value dimensions expressed, and that participation—although considered key—rarely reports on inclusivity. The findings suggest that in Flanders, a greater emphasis is placed on relational values associated with urban nature. A broader value scope for the design, management, and evaluation of urban green spaces tailored to the local context is recommended.

**Keywords:** relational values; plural valuation; nature-based solutions; urban nature; urban green spaces

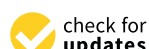



## 1. Introduction

More than half of the world's population currently lives in urban areas, and this number is projected to reach 70% by 2070 [1]. Cities consume over 78% of total human resources and are responsible for 70% of greenhouse gas emissions, yet they occupy less than 2% of the Earth's surface [2,3]. Furthermore, environmental crises such as climate change exacerbate drivers of urban inequality, threatening the future of cities [3]. While cities are centers of economic, social, and cultural innovation, they face significant challenges due to population growth outpacing adaptive capacity [4].

These challenges include biodiversity loss, climate change impacts, health issues, and social injustices. Cities are highly susceptible to inequalities, leading to higher crime rates, social exclusion, and income disparities [5]. Additionally, policy proposals that offer singular approaches to urban problems have been found to have unintended negative consequences, such as green gentrification [6]. This causes the displacement of low-income residents because of environmentally conscious urban development projects [7]. In many urban areas, wealthy neighborhoods tend to have better living conditions, while less

affluent areas often experience poorer conditions, leading to negative health outcomes [8]. This means that social inequality can cause groups of people in urban areas to face greater health risks due to factors such as air pollution and heat stress [9–11].

This will increasingly be an issue because urban areas are warming at double the rate as their rural counterparts [12,13]. As cities become denser and expand, the amount of green area in and around them decreases, leading to a significant contribution to the urban heat island effect [14–16]. In the future, climate change consequences will pose an increasing danger to urban citizens, including more frequent and severe heat waves, storms, and droughts, and rising sea levels [12].

The other environmental challenge cities are facing is their negative impact on biodiversity. The homogenization of species, fragmentation, and loss of open, green space are all contributing factors to the decline in biodiversity in urban areas [17]. Moreover, the level of green area in a city is a crucial determinant of its biodiversity, with a drop in biodiversity occurring when the percentage of green area drops below ten percent [18]. As cities continue to expand, striking a balance between urban development and green areas becomes increasingly difficult [16].

This range of urban challenges can effectively be addressed by nature-based solutions (NBS). NBS draw inspiration and power from nature and have proven to be effective in tackling societal challenges, promoting biodiversity, and providing multiple benefits [19]. NBS are projects that deliberately work with ecosystems to ensure additional benefits for people and nature in comparison to grey infrastructure [19]. These projects can vary in their implementation, from using to restoring ecosystems or to even creating new ecosystems [20]. The multitude of benefits provided by NBS at a relatively low cost makes them suited in the response to many urban challenges [21–23]. NBS are also seen as a way to promote sustainability and equity in cities [23]. A comprehensive approach that incorporates NBS as part of the solution can address the aforementioned challenges in many ways including but not limited to the following.

NBS can contribute to the easing of social issues in cities, e.g., the presence of parks and urban green spaces reduces crime and can promote social cohesion [24,25]. By facilitating social cohesion and physical activity and reducing environmental stressors and pollutants these urban green areas also influence residents' health [26,27]. Furthermore, people living near urban green have been found to feel healthier and happier [9,27–29]. In the face of climate change and increasing urban heat island effects, parks tend to be one degree Celsius cooler than the surrounding urban areas and this cooling effect extends beyond their borders [30,31]. Finally, NBS have emerged offering a way to create green infrastructure and enhance urban biodiversity [32]. Cities can be hotspots for threatened species; thus, by creating habitats urban biodiversity can increase, even in tiny patches [33,34].

The focus of this article is on NBS implemented in cities in Flanders, a region in Belgium that has experienced significant urbanization since the 1950s [35]. With one-third of its territory occupied by settlements, Flanders has become Europe's most urbanized region [36,37]. Impervious surfaces cover 14.93% of Flanders, with 60% of this surface not occupied by buildings [38]. Compared to other wealthy and densely populated European regions of similar size, Flanders has the highest share of urban and built-up areas and the lowest share of natural ecosystems [36,37]. Consequently, Flanders is confronted with the urbanization challenges mentioned previously.

Flanders is experiencing a growing demand for urban green spaces, which are diminishing in size and unable to keep up with the increasing population dependent on them [38–40]. A governmental report published in 2012 has outlined various social benefits associated with urban green in Flanders, for instance, they function as a catalyst for social cohesion and integration, as well as foster a sense of community and belonging among residents in their respective neighborhoods [41].

Among the challenges that Flanders is currently facing in urban areas, one of them is related to threats to the health of its citizens. Air quality is a major concern in Flanders, with WHO standards being exceeded everywhere, resulting in 4200 premature deaths due to

particulate matter in 2021 [42]. A recent literature review by the Belgian health department found direct links between the presence of urban green spaces and health benefits in Flanders [43]. Another major concern is heat stress in urban areas. During recent summers, there was a significant amount of excess mortality during summer heatwaves [38,44].

The impacts of climate change are increasingly visible in Flanders, where heavy rainfall days have been on the rise [45]. Heat waves are expected to become more frequent, longer, and hotter in the coming years [44,46]. Flanders has already experienced a warming of 2.46 °C since the 19th century, which is faster than the global and European averages [46]. This increased warming trend is largely attributed to land cover changes, particularly from urbanization [12,14,47]. Flanders' 2050 climate strategy includes a section on climate adaptation, which highlights the use of NBS as a means of adapting to the effects of climate change in urban areas. The strategy recommends the use of NBS to address the aforementioned urban challenges [47].

This would be beneficial for biodiversity in Flanders as well. The most recent nature report from the Flemish Research Institute for Nature and Forests highlights that increased urbanization is fragmenting natural areas in Flanders, which is a significant pressure on biodiversity. The report notes that 89% of the natural areas are smaller than 1 ha, resulting in edge effects and limited dispersal opportunities. In urban areas, less than half of the species found in undisturbed scenarios are present. To address this issue, the report suggests increasing the presence of green–blue infrastructure in urban areas, which could improve the ecological quality and connectivity of the urban environment [37].

Assessing urban nature involves many variables, as green spaces play a vital role in offering diverse functions to local inhabitants and the cityscape. As a result, research in this field requires different disciplinary perspectives, such as biodiversity science, and social and economic sciences. However, this study aims to provide an overview of the diversity of functionalities and values that urban nature provides. A plural valuation framework is applied to achieve a broad perspective [48–52].

The framework groups the diverse values associated with urban green space into instrumental, intrinsic, and relational values. Instrumental values refer to nature's benefits to humans, associated with nature as an asset or resource, while intrinsic values pertain to the worth of nature independent of any reference to humans as valuers and are worth protecting for their inherent value. Relational values describe the significance of interactions between people and nature and interactions between people through nature, including a sense of place, spirituality, care, and reciprocity.

This study provides the first plural values-based assessment of NBS communication in Flanders, based on a large and diverse sample of project descriptions. It provides an elicitation of the diverse values of communication beyond disciplinary perspectives from either biodiversity, ecosystem services, economic or social literature, and provides a solid and legitimate basis for future research and policy regarding the design and evaluation of such projects. It offers valuable insight, which is currently lacking, into what real-world practitioners put forward in their communication.

The aim of the research is to uncover the values linked with NBS undertakings in Flanders. The goal is to identify which particular values are highly valued in Flanders and to explore potential links between these communicated values and the traits of the locality and the projects carried out. The first hypothesis is that the different value dimensions will be present in varying degrees. Additionally, it is hypothesized that a broad suite of values will be important regardless of the project and its background. Furthermore, larger projects are expected to use more diverse values.

## 2. Materials and Methods

In this section, the methods for investigating the values found in the descriptions of nature-based solutions in Flanders are described. A total of 106 projects were gathered across twelve cities, including park restructurings, street redesigns, and housing developments. The analyzed projects were evaluated using a code tree that had three dimensions:

intrinsic values, instrumental values, and relational values. Additionally, data on the socio-economic parameters of the surrounding area of the projects were collected, and descriptors were developed for further analysis. The coding process aimed to assign the most specific code to statements. The collected data were then used to explore possible connections between socio-economic and environmental factors and the values communicated.

### 2.1. Data Sources

The projects originated from the three main online accessible repositories on these projects in the Flanders region. These three repositories together offer the widest range of NBS-type projects in Flanders. All repositories, although slightly varying in focus and embedded in various parts of the Flanders administration, have the common goal of inspiring professionals and the general public (Table 1).

**Table 1.** Characteristics of the repositories.

| Repositories | "Openbaar Groen" | "Vlaanderen Breekt Uit" | "Blauwgroen Vlaanderen" |
|---|---|---|---|
| Organization | "VLAM": Flemish center for agriculture and fishery marketing "VVOG": Flemish association formed around public green spaces | Flemish department of the Environment | "VLARIO": Water management knowledge center "Aquafin": Flemish water agency |
| Goal and Audience | Inspiring the public and local policymakers with exemplary green space projects | Inspiring citizens and local governments to start their own desealing projects | Inspiring public space planners to design climate- and nature-friendly projects |
| Information Gathering | Submission of dossiers for the selection of an annual award | Project and funding solicitations, (initiatives for desealing pilot schemes, . . . ) | Application form for stakeholders who want to highlight projects |
| Number of cases on website | 174 | 57 | 84 |
| Number of cases selected [1] | 64 | 15 | 29 |
| URL | https://www.openbaargroen.be/projecten (accessed on 12 November 2022) | https://omgeving.vlaanderen.be/nl/realisaties (accessed on 12 November 2022) | https://blauwgroenvlaanderen.be/professionals/projecten/ (accessed on 12 November 2022) |

[1] Totaling 108 rather than 106 is due to some projects being recorded in two repositories.

### 2.2. Case Selection

Included projects met two criteria:

(1) They were under construction or completed at the time of the selection process, October 2022;

(2) They were located in a Flemish city classified as a "main city" ("centrumstad", Figure 1), which entails having a population of at least 80,000 people, a significant social and economic impact, and meeting various geographical criteria [53,54]. As a result, projects are highly diverse regarding their socio-economic and physical characteristics. The final sample of 108 descriptions of 106 projects can be regarded as representative of the diversity of successful green space projects in the main urban areas in Flanders.

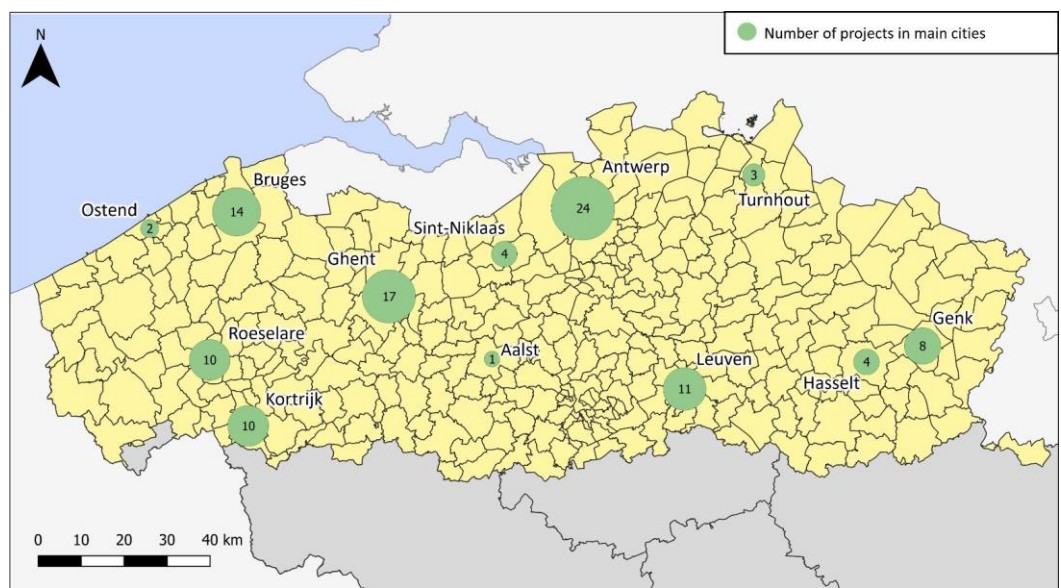

**Figure 1.** Map illustrating the distribution of sampled projects. 106 projects: Antwerp, 24; Ghent, 17; Bruges, 14; Leuven, 11; Roeselare, 10; Kortrijk, 10; Genk, 8; Hasselt, 4; Turnhout, 3; Ostend, 2; Aalst, 1.

*2.3. Theoretical Framework: Code Tree*

Once criteria for inclusion had been established, the analysis of selected projects' online descriptions commenced using a code tree. This entails a hierarchical classification system of codes applied to textual information. The objective was to identify any mentioned benefits and values.

This code tree is based on the specific values framework of IPBES [51]. The structure of the code classification was adopted from the URBAN Gaia study regarding indicators of urban green space [48]. The dimensions of the code tree can be aligned with the IPBES values.

- **"Nature" or intrinsic values:** Worth of nature independent of any reference to humans as valuers and are worth protecting for their inherent value.
- **"Nature contributions to people" (NCP) or instrumental values:** Nature's benefits to humans, associated with nature as an asset or resource.
- **"People" or relational values:** Importance of interactions between individuals and nature, as well as between individuals through nature, such as a connection to place, spiritual beliefs, acts of caring, and mutual exchange.

Throughout the research process, additional adjustments were made to the code tree. These modifications encompassed new aspects and categories and served to enhance the assessment of urban green initiatives (see results Section 3.1 for the final code tree). Adaptations to the code tree were made in discussion between authors, based upon issues uncovered during the coding process.

*2.4. Elicitation of Values: Content Coding*

After the selection and text extraction, each sentence or statement was coded using Dedoose software following the adapted code tree [55]. Statements were coded using the most specific code available in the code tree. Only statements explicitly mentioning a certain value, e.g., "this measure was taken to improve local biodiversity," were coded to reduce personal interpretation bias, improve traceability, and ensure repeatability. In case of doubt or disagreement on coding of certain statements, final codes were assigned after deliberation between the authors. This resulted in 567 coded statements representing 977 communicated values.

## 2.5. Descriptors

In parallel with the elicitation of values, physical, ecological, and social descriptors of each project and surrounding context were gathered. Correlations between these socio-economic and environmental variables and the elicited values were explored. To achieve this objective, data pertaining to the socio-economic parameters of the surrounding neighborhoods were sourced from a public database [56]. Pearson correlation analyses were conducted to explore relationships among parameters related to ethnicity, wealth, and spatial context. Strongly correlated parameter groups were identified. To streamline the analysis and avoid redundancy, an indicator was selected by prioritizing interpretability (Table 2). The chosen parameter demonstrated strong correlation within its group and offered straightforward interpretation in the study context, e.g., green area was chosen to be more informative than non-build up area.

**Table 2.** Definitions of the socio-economic and environmental descriptors.

| Descriptor | Description |
| --- | --- |
| Population density | Number of inhabitants per square km |
| Percentage of inhabitants with non-Belgian heritage | Percentage of inhabitants in the neighborhood with at least one parent born abroad |
| Green area | Percentage of green space in the neighborhood |
| Interquartile coefficient | Ratio of the income difference between the third and first quartiles, adjusted for inflation by dividing by the median income, indicating income inequality |
| Interquartile asymmetric | Measure of income distribution asymmetry, with a higher number indicating greater concentration of high incomes above the median |
| Average net taxable income | Average net taxable income of the neighborhood |

Furthermore, an indicator was developed to assess the degree of technical and natural measures implemented in every project. A nature gradient of the Flanders nature report was utilized for this purpose [57]. This concerns a 5-step scale ranging from completely technical (e.g. parking lot renewal with water permeable paving) to completely natural measures (e.g. creation of novel wetland as a water buffering area), with intermediate steps including dominant technical with a natural presence (e.g. square renovation with trees and grass as cooling elements), equal measures (e. g. green roof constructions), and dominant natural measures with a technical presence (e.g. green swale connected by tubes) (Table 3) in order to verify if a pattern emerges.

**Table 3.** Nature–technical gradient steps explained with descriptions, examples, and pictures.

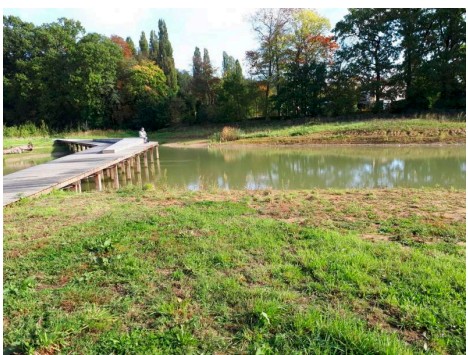

completely natural, Park Ten Rozen, Aalst

**Table 3.** *Cont.*

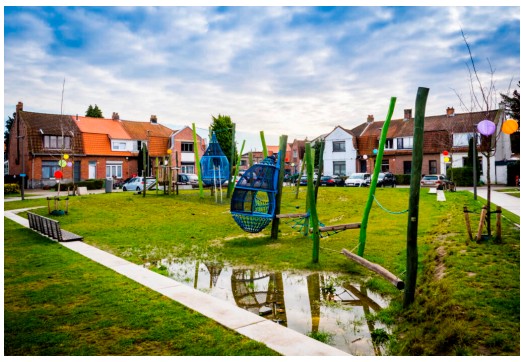

dominant natural with technical presence, Florent Cootsmanplein, Antwerp

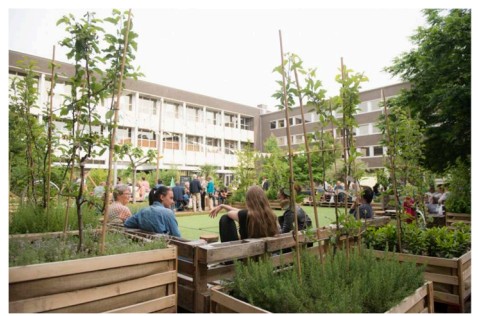

equal measures, Pop-up Tuin, Genk

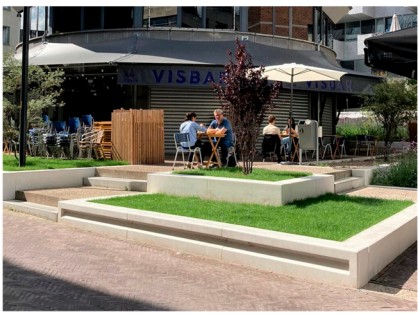

Alfons Smetsplein, Leuven

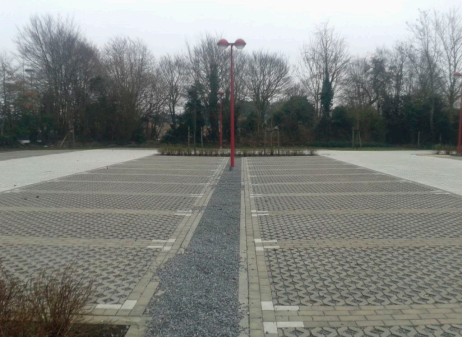

completely technical, Parking kerkplein, Hulshout.

Two researchers independently evaluated each project, and in cases where there was disagreement, a third researcher was consulted to resolve the discrepancy in deliberation. The researchers mainly used the project measures but also green and grey elements in project images, satellite imagery, and Google Street View as criteria.

A "size descriptor" for each project was also developed, which was based roughly on the Flemish green space standards. [58]:

- **Local**: street or small neighborhood green areas.
- **Small**: park-sized projects.
- **Large**: neighborhood-sized projects, housing development.

Subsequently, the participation policy ladder developed by Edelenbos in 2000 was used to indicate the participation level in each project [59].

- **Informing**: This involves one-way communication from the project organizer to the local public.
- **Consultation**: This level allows the local public to voice their opinions without any commitment from the organizer to take them into account.
- **Advising**: At this level, the local public plays an active role and provides feedback that the organizer takes into consideration. With substantial justification, it remains possible to reject input from the public.
- **Co-producing**: This level involves a strong commitment from the local public from start to finish, resulting in a clear impact of their involvement, only limited by predefined conditions.
- **Co-deciding**: At the highest level, the local public takes the initiative for the project and leads it from start to finish, with an advisory role for policymakers.

For analysis purposes, the participation levels were consolidated into two categories: "passive" (which involves informing and consultation) and "active" (which involves advising, co-producing, and co-deciding). This categorization aligns is based upon Arnstein (1969) [60].

Additionally, an investigation was conducted to determine whether any actions were taken to enhance inclusion in the participation process. A parameter was then established to reflect the presence or absence of inclusion measures, and another parameter to specify the type of measure taken, such as providing multiple participation opportunities, establishing a comprehensive participation project, or targeting specific audiences such as youth or the elderly. The results of these descriptors can be found in Tables A1 and A2 in Appendix A.

*2.6. Multifunctionality*

In order to reach a conclusive analysis, a numerical indicator was constructed to encompass the diversity of dimensions, categories, and subcategories into a single index. This index is higher when subcategories or categories from different dimensions are co-occurring. This multifunctionality index (see also Hölting et al. [61]) is calculated in the following way:

$$MF_P = D_P \times 3 + C_P \times 2 + SC_P \times 1, \tag{1}$$

$D_P$: Dimensions present at project p
$C_P$: Categories present at project p
$SC_P$: Subcategories present at project p
3: Dimensional factor
2: Category factor
1: Subcategory factor
Example below:
Victoria Regia Park in Ghent:

- Two dimensions present ("NCP" and "People") = 2 × 3 = **6**
- Three categories present ("Non-material services", "Cultural" and "Health & Wellbeing") = 6 + 3 × 2 = **12**
- Six subcategories present ("Experiences", "Heritage values", "Identity, sense of place", "Stewardship", "Education & Knowledge" and "Social relations") = 12 + 6 × 1 = **18**

*2.7. Analysis*

A co-occurrence analysis of the subcategories was conducted using the "cooccur" R package [62]. Only subcategories that appeared more than fifteen times were chosen to ensure clarity of the analysis. The "Eulerr" R package was utilized to visualize the co-occurrence of the value dimensions [63].

Multiple statistical analyses were conducted to ensure the robustness of the findings. An ANOVA test was conducted to compare the variance in the socio-economic descriptors

grouped by the presence of a subcategory, category, or dimension. This test elicits if any values are correlated with the context of the neighborhood. Another ANOVA test was performed with the socio-economic descriptors grouped by the different project descriptors. Additionally, a linear regression was carried out between the inhabitants of each city and the number of projects.

## 3. Results

### 3.1. Framework Presentation and Adaptation Process

As previously stated, the starting point for the research was the IPBES-based KPI framework developed by Carmen et al. (2020) [48]. Several modifications were made throughout the research process, leading to the presentation of the final code tree in Table 4.

**Table 4.** Finalized code tree used for the coding.

| Dimension | Category | Subcategory |
|---|---|---|
| Nature | | Biodiversity |
| | | Ecological connectivity |
| | | Biophysical processes |
| | | Individual organisms |
| | | Nature itself (green space) |
| NCP | Material contributions | Energy, food and feed, materials, medicinal, biochemical, and genetic resources |
| | Regulatory contributions | Regulation of air quality |
| | | Regulation of local climate |
| | | Regulation of global climate |
| | | Regulation of hazards and extreme events |
| | | Regulation of freshwater quality, flow, and timing |
| | | Habitat creation and maintenance |
| | | Pollinators and dispersal of seeds |
| | | Formation, permeability, and decontamination of soils |
| | Non-material contributions | Physical and psychological experiences |
| People | Cultural | Heritage values |
| | | Identity, sense of place |
| | | Stewardship |
| | Economy | City attractiveness |
| | | Cost-efficiency and robustness |
| | | Jobs |
| | | Profits for business |
| | | Property values |
| | Governance and justice | Distributional justice |
| | | Procedural justice |
| | Health and wellbeing | Education and knowledge |
| | | Physical and mental health |
| | | Safety and security |
| | | Social relations |
| | Mobility | Reachability |
| | | Connectivity of paths and roads |
| | | Accessibility |

- "Quantity and quality of GBI": This category was moved to the "People" dimension and renamed to "Mobility". Most GBI subcategories were combined into two categories: "Connectivity of paths & roads" and "Accessibility". These categories represent a project's mobility and infrastructure. Next to this, the added subcategory "Reachability" covers how easy it is to reach the project and the mobility issues or solutions that come with it. This category fits better in the relational values dimension than in the intrinsic value dimension.

- "Economy" category: A new subcategory named "Cost-efficiency and robustness" has been included, which encompasses budgetary incentives associated with these projects, including expenses during construction and planning as well as future costs. This includes adaptations to climate change.
- "Non-material contributions" category: The subcategory "Supporting identities" was removed since it was adequately covered by the subcategories "Experiences" and "Identity, sense of place". The dotted line between "Experiences" and "Identity, sense of place" is meant to illustrate the close link between both subcategories. The interchangeability between both can certainly be argued and this is a potential future adaptation.
- "Material contributions" category: All subcategories were merged into one because of the limited need for these different subcategories.
- "Regulation of climate": This subcategory was split into "Regulation of local climate" and "Regulation of global climate". The former includes local temperature control measures, for instance, the placing of trees to mitigate the urban heat island effect. The latter is mainly found in climate mitigation measures pointing at a reduction in $CO_2$ emissions. This was found to be more representative when separated.
- "Regulation of ocean acidification" and "Regulation of organisms posing harm to humans": These subcategories have been dropped from the code tree. Both did not occur in this research. Future research might find these subcategories relevant to their focus.

### 3.2. Descriptor Results

The results of the project descriptors will first be presented. A summary of the levels of participation, inclusivity measures, and the range of measures on the natural–technical scale, along with the distribution of project sizes within the sample, is available in Table A1 (Appendix A).

A considerable proportion of projects did not report any participation or inclusion measures. A majority of the projects that did report a participation event chose an active form of participation and handed some influence on the public. When inclusion measures were taken, a notable number of projects set up an extensive participation project with significant efforts for including diverse types of residents.

There were no projects that took solely technical measures. Almost half of the project implemented mainly nature-based measures. Additionally, it is worth noting that the descriptor "local", which is the smallest in size, is the most frequently used one.

The socio-economic backgrounds of these projects were found to be varying along the six selected parameters. These results illustrate that the data sample covers a broad range of different urban areas. The variety is clear in terms of population density, non-Belgian heritage, green area, and income (inequality) parameters, as illustrated in Table A2 (Appendix A).

### 3.3. Overview of Values Distribution

Recurrently, the analysis revealed that the distribution pattern consistently reflected roughly 55% for "People", 30% for "NCP", and 15% for "Nature". This ratio is present at the aggregate level encompassing all data (Figure 2) but also across all the descriptors analyzed, including size, natural–technical scale, participation and inclusion, and repository. It is also notable that the "Nature" dimension is sustained by both "Biodiversity" and "Nature itself". Similarly, in the "NCP" dimension, "Regulatory contributions" are highly valued and there is no single subcategory that dominates. "Material contributions" did not have much impact, but "Experiences" form a large part of the "NCP" mentions. The "People" dimension was spread out evenly among the categories with varying importance of subcategories.

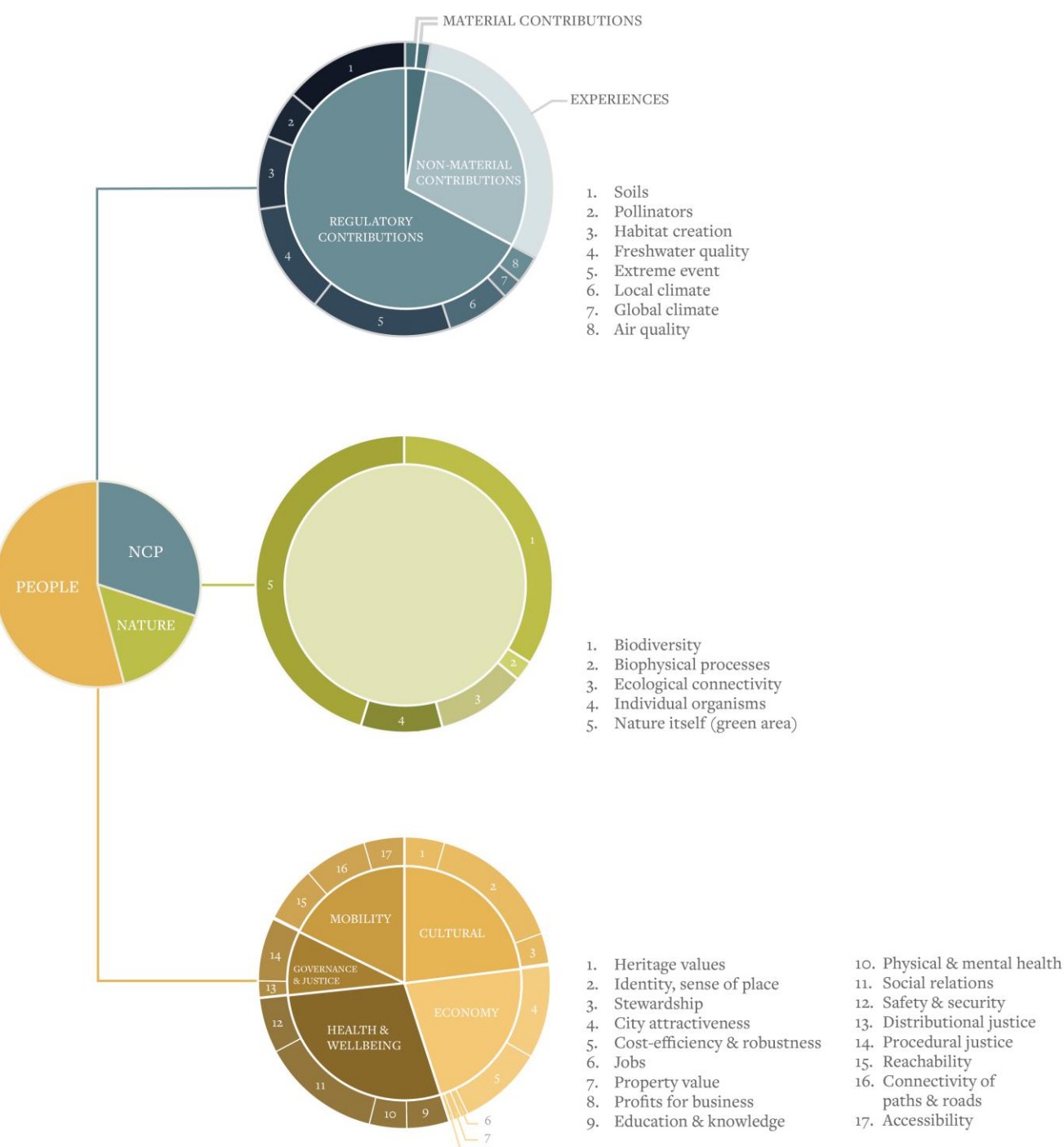

**Figure 2.** Relative distribution of value dimensions, categories, and subcategories as coded from statements (n = 567) from 108 descriptions from public repositories about urban green projects in main cities in the Flanders region.

Another group of subcategories that occur more frequently with each other is focused on well-being and interactions, which includes "Experiences", "Nature itself", "Social relations", "Identity, sense of place" and "Safety & security".

*3.4. Co-Occurrence*

3.4.1. Co-Occurrence of Dimensions

The co-occurrence of the dimensions shows that the most common combination features all three dimensions (Figure 3). Additionally, the findings reveal that the "People" dimension is ubiquitous, as only a small number of projects (i.e., five) do not incorporate "People" values. Note that one project in the sample did not use any value of the code tree, containing strictly technical information.

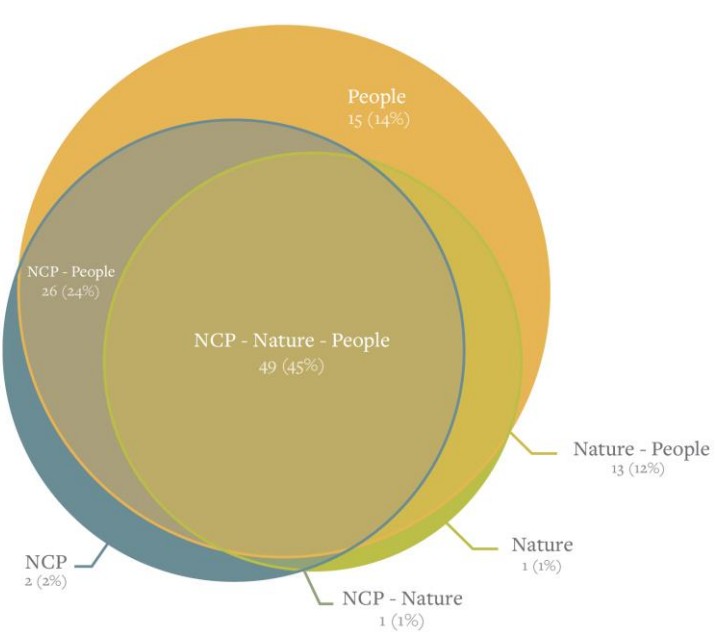

**Figure 3.** Co-occurrence of value dimensions. The figure shows how the combinations relate to each other, but the surface area ratios are not accurate.

3.4.2. (Co-)Occurrence of Subcategories

When looking at the individual subcategories, the results show that the "Experiences" and "Identity, sense of place" subcategories are the most commonly used values in these projects (Figure 4). Additionally, the subcategories of "Biodiversity" and "Nature itself" occur quite frequently when compared to other subcategories. The "People" dimension is clearly most prevalent as seen in Figure 2. When examining infrequently utilized subcategories, monetary-focused ones such as "Property value," "Business profits," and "Employment" can be observed.

Notably, not a single subcategory is present in half of the projects, indicating a diversity of values used across the projects. These findings demonstrate a consistency in the dimensions of values, but not in the specific subcategories used in different projects.

The co-occurrence analysis of the subcategories shows a group of subcategories frequently occurring together that pertain to the management of water, including "Regulation of extreme events", "Regulation of soils", and "Regulation of freshwater quality" in combination with "Biodiversity" and "Nature itself" (Figure 5).

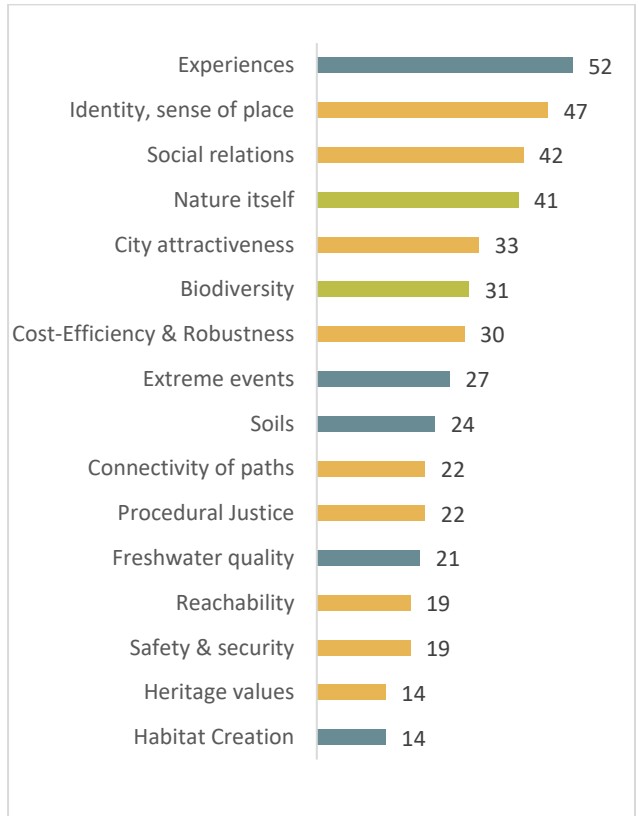

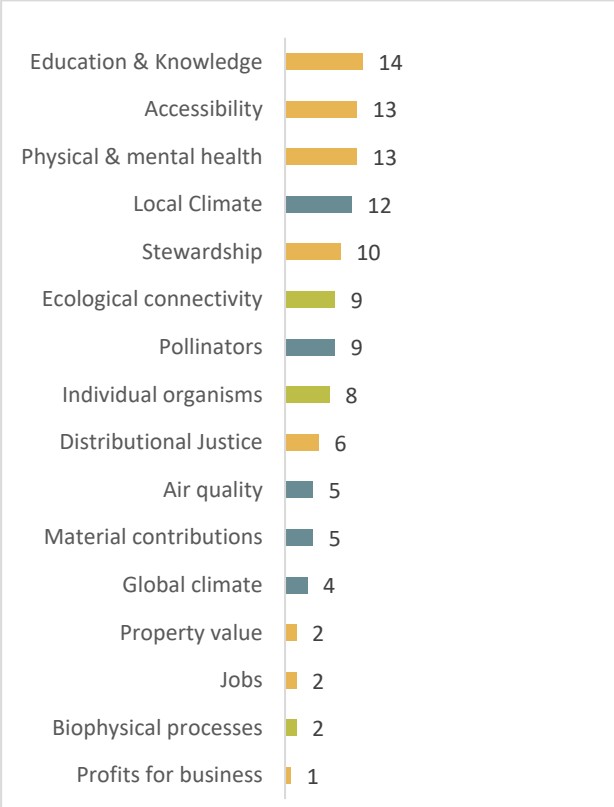

**Figure 4.** Total occurrences of subcategories (n = 573) as coded from statements from 108 descriptions from public repositories about urban green projects in main cities in the Flanders region.

SUBCATEGORY CO-OCCURENCE MATRIX

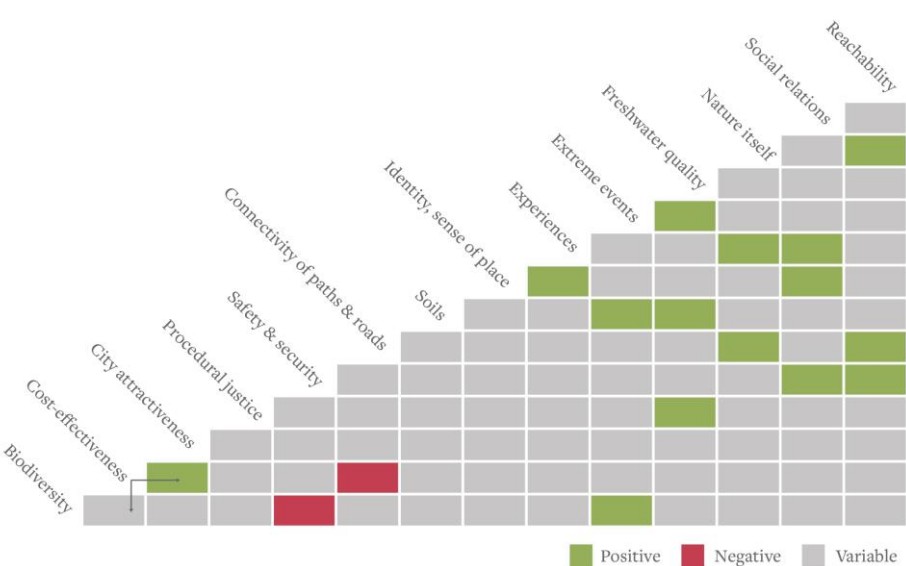

**Figure 5.** Co-occurrence matrix of the subcategories that were present more than fifteen times in the dataset. Positive results indicate that these two subcategories occur more frequently together than expected by chance.

### 3.4.3. Multifunctionality Descriptor

This study used a multifunctionality index, which was developed as explained in the Section 2. Based on this descriptor, it has been found that multifunctional communication

varies across the sample. A majority of the projects use a moderate level of value variety in their communication. The remaining projects are split equally between those that report high and low levels of multifunctionality (Table A3 Appendix A).

Furthermore, an analysis was conducted which revealed a notable positive correlation between project size and multifunctionality (Figure 6, *p*-value = 0.000183, Table A4), with larger projects displaying higher levels of multifunctionality compared to their local counterparts. This correlation was also evident with respect to participation, inclusion, and the type of intervention measures, all of which yielded significant analysis results (participation *p*-value = 0.0177, inclusion *p*-value 0.000116, nature-based intervention scale *p*-value = 0.0026, Table A4, Appendix A). Conclusively, larger projects, or projects with more participatory, inclusive, or nature-based measures tend to score higher on the multifunctionality index.

**Figure 6.** Boxplots of the multifunctionality index grouped by the project size descriptor.

*3.5. Statistical Analyses*

Most of the statistical analyses performed were not significant except for the significant, positive linear regression (*p*-value = 0.000759, adj $R^2$ = 0.664, Table A4, Appendix A) that was observed between the number of reported projects in a city and its population size. Furthermore, the ANOVA results for socio-economic descriptors and value dimensions were insignificant, revealing that there is no implicit bias in communication about the projects related to ethnic make-up, wealth, or population density of the neighborhood.

**4. Discussion**

This study has several limitations that should be acknowledged. Firstly, it is essential to emphasize that this study relies on statements found in project descriptions within the framework of inspiring communication. This does not necessarily reflect whether these are ambitions or actual achievements. However, these statements could be seen as claims which could be a departure point for future performance evaluations. Secondly, there is a bias in the project sample. These repositories only display successful projects that are highlighted by the responsible parties. However, they do not provide reporting on underappreciated or failed projects. It is an important consideration for future research to look for similar projects that are not as highlighted. A central repository that compiles these NBS projects for the entire region, regardless of success or funding, would be beneficial for future research.

### 4.1. Consistent Abundance Ratios of Value Dimensions

While values associated with urban green projects are highly diverse, the main finding is that the distribution of dimensions remains remarkably constant across different repositories, contexts, and project types. The dimension related to "People" consistently ends up between 50–60%, with "NCP" at 20–30%, and "Nature" at 10–20% (Figure 2). It is important to note that these values are explicitly mentioned in the projects' description in order to be coded as such and that they are the values that the project managers seek to highlight.

One hypothesis is that this distribution is driven by the number of subcategories per dimension (17/32~53%, 10/32~30%, and 5/32~15%). However, the occurrence of the single subcategories clearly shows that a small number of subcategories strongly influence the dimension distribution, rather than an equal distribution over subcategories per dimension (Figures 2 and 4). Moreover, when disregarding a substantial part of the data and selecting the same number of subcategories (five most abundant) for each dimension, relational values are still the most prevalent dimension (43% "People", 34% "NCP" and 23% "Nature"). So, regardless of the potential influence of the coding framework, relational aspects seem to be consistently the most abundant—and thus important—values associated with urban green infrastructure in Flanders.

These relational values are a prominent feature in the IPBES values assessment and similar literature. While research has been focused on instrumental and intrinsic values, a growing call has been emerging to give equal attention to relational values which resonate broadly and differently [51,64,65]. The results show that relational values are abundantly present. The main relational values highlighted are the emotions or experiences that people will have when they visit a particular site. The three most frequently occurring values in this regard are "Experiences", "Identity, sense of place", and "Social relations" and they often occur together (Figures 4 and 5). Additionally, there is a significant emphasis on the intrinsic value of nature and its beauty, evident in the values of "Nature itself", "City attractiveness", and "Biodiversity". However, purely economic values are noticeably absent from the descriptions, suggesting they are less important here.

Prior research has emphasized the importance of relational values in personal preferences. For instance, a study in Australia investigated people's preferences for parks and determined that the most desired values were associated with health and safety [66]. Similarly, Arias-Arévalo et al. (2017) conducted a study on individuals' preferences for a river system and discovered that relational values were ubiquitous in people's preferences [67]. Drawing from these value distributions, it is evident that relational values are assuming a pivotal position in people's perception of these NBS projects. Thus, this research confirms that there might be a mismatch between the values most often the focus of research, and the values regarded as important in practice [51].

Previous research has found that personal variables (e.g., age and gender) can influence preferences for urban green spaces [68]. Salm et al. (2023) found that income and the amount of green in the neighborhood can also influence urban green preferences, indicating that social or environmental variables may have an influence as well [69]. This study did not focus on differences in personal preferences. However, the high variability in subcategories might point to the impact of local context but average neighborhood income and greenness did not impact the values communicated. It is thus possible that the communicated values in the public repositories do not fully reflect the diversity of preferences between social groups.

Examining the co-occurrence of the subcategories (Figure 5), a clear group emerges. There is a group of subcategories that are explicitly mentioned in the context of water management ("Extreme events", "Freshwater quality", "Soils"). The recent literature has confirmed the effectiveness of NBS for urban water management [70,71]. Oral et al. (2020) highlight the benefits of restoring water to the natural hydrological cycle, which is the focus of the desealing projects in the data sample [70]. Huang et al. (2020) clearly state that this helps greatly with water quality and run-off management [71]. This is confirmed by

these results as being an important focus of urban NBS in Flanders through which project managers want to illustrate the novel, nature-based approach to responding to increasing heavy rainfall.

Participation is often considered crucial for taking into account the diverse needs and interests of citizens in the context of NBS [19,23,72]. Studies have shown that participatory efforts are strongly associated with positive social sustainability outcomes, such as social learning, a sense of belonging, and environmental stewardship [72]. This research also examined the correlation between reported participation and the distribution of the values. Notably, this study found a substantial number of reported citizen-powered participatory efforts, which is not frequently found in the literature [73]. While no clear correlation was found between participation and the distribution of values, it should be noted that almost half of the analyzed projects did not report on their participatory efforts, leaving room for uncertainty. It is possible that these projects did involve participants and therefore reported the values preferred by the community, or conversely, that they were not inclusive and did not align with local needs and desires. Thus, this remains a blind spot, and further research is needed to draw any definitive conclusions.

*4.2. Multifunctionality*

In this study, a multifunctionality index was adopted to quantify the diversity of values reported. This analysis revealed a high diversity of values and a significant correlation observed between project size and multifunctionality. This is consistent with the expectation that larger projects have more resources, surface, and flexibility to pursue a variety of approaches to promote their goals (Figure 6).

By examining the distribution of dimensions (Figure 3) and the multifunctionality index (Figure 6), it becomes evident that there is a strong tendency to incorporate multiple values into project representation. Nature-based solutions are not just a nature reserve, not just a social space or water regulation system, they are often all these combined.

However, a point to be made is that highly diverse functions are not necessary for every project. Hölting et al. (2019), argue that a mono-functional project can still be valuable by providing a unique value to the neighborhood, called beta-multifunctionality [61]. In these cases, a project can be especially valuable if it offers services that are missing in that area. For example, a small open grass field in the city may not offer diverse benefits, but it can still be highly valued because it might be the only open space in the neighborhood.

## 5. Conclusions

Circling back to the hypotheses, the conclusion is these dimensions were indeed important in these descriptions, with a clear dominance of relational values. The statistical results showed that there was no correlation between the values used and the parameters of the project or neighborhood. There was a positive correlation between project size and the diversity of values used.

The inventory of diverse values associated with urban green spaces in Flanders can form a legitimate basis for research, design, management, and evaluation of NBS projects in Flanders. It offers a legitimate, practice-based picture for practitioners on what values are appreciated by society, what values to consider during the process, and what other communicators use to describe their projects. The code tree can prove to be particularly beneficial during the initial stages of project visualization and participation, as it helps in directing the project's objectives and intentions toward societal priorities. Moreover, this value framework can provide a start for the evaluation of these projects.

The findings demonstrate both the diversity of specific values prioritized in urban green projects and the consistent emphasis on relational values as the largest value dimension. Two practical recommendations can be made based on this: (1) the value code tree is not universally applicable and should be used as a starting point, adapting to the local social and physical context, and (2) more attention is needed for relational values, how to quantify, qualify and evaluate them regarding NBS.

This research is the first to explore values in urban nature projects and emphasizes the importance of relational values, often overshadowed by biodiversity and economic indicators [51]. Relational values need to be considered by researchers when looking at the value of urban nature. These values might be hard to quantify, monetize, or visualize but they clearly are relevant to research into urban NBS projects.

**Author Contributions:** Conceptualization, T.B., E.V.W. and S.J.; methodology, T.B., E.V.W. and S.J.; software, T.B.; validation, T.B., E.V.W., S.J. and C.E.B.; formal analysis, T.B; data curation, T.B.; writing—original draft preparation, T.B.; writing—review and editing, S.J., E.V.W., J.D. and C.N.; visualization, T.B., K.K. and L.V.; supervision, S.J. All authors have read and agreed to the published version of the manuscript.

**Funding:** This research received no external funding.

**Data Availability Statement:** The descriptions used for this study are available at the URLs provided in Table 1.

**Acknowledgments:** We are grateful for the broad overview delivered by these online repositories. We would like to thank Aquafin and VLARIO, Department Omgeving, and VLAM and VVOG for creating these initiatives and making this research possible.

**Conflicts of Interest:** The authors declare no conflict of interest.

**Appendix A**

Table A1 shows the project descriptors distribution used in the study. The "Natural-technical measures scale" indicates the degree to which natural and technical measures were used in each project, ranging from completely technical to completely natural. The "Size descriptor" describes the size of each project, based on the Flemish green space standards [58]. The "Participation" indicates the level of public involvement in each project, categorized as passive or active, and as the participation policy ladder developed by Edelenbos [59]. Finally, the "Inclusion Measures" specifies whether any measures were taken to increase inclusion in the participation process, and if so, the type of measure taken.

**Table A1.** Project descriptor results.

| Participation | | | | | |
|---|---|---|---|---|---|
| **Passive** | | | **Active** | | **Not reported** |
| 24 (22%) | | | 32 (30%) | | 52 (48%) |
| Informing | Consultation | Advising | Co-produce | **Co-decide** | **Not reported** |
| 12 (11%) | 12 (11%) | 11 (10%) | 16 (15%) | 5 (5%) | 52 (48%) |
| Natural–technical scale for measures | | | | | |
| Completely natural | Dominant natural with technical presence | Equal measures | Dominant technical with natural presence | **Completely technical** | |
| 7 (6%) | 44 (41%) | 36 (33%) | 21 (19%) | 0 (0%) | |
| Inclusion | | | | | |
| Multiple participation moments | Participation project | Specific target audience measures | Not reported | | |
| 11 (10%) | 13 (12%) | 4 (4%) | 80 (74%) | | |
| Size | | | | | |
| Large | Small | Local | | | |
| 17(16%) | 27 (25%) | 64 (59%) | | | |

Table A2 in the appendix of the article presents the results of the socio-economic data of the areas surrounding the projects. These data were sourced from the public database and included information on eighteen distinct parameters, which were analyzed

through Pearson correlation analyses to eliminate similar parameters. The six most relevant parameters were selected and used as descriptors in Table A2. These parameters provide information on the socio-economic context of the projects and help to understand how the projects relate to the surrounding neighborhoods.

**Table A2.** Socio-economic descriptors results.

| Percentage of inhabitants with non-Belgian heritage | | | | |
|---|---|---|---|---|
| (0.63–15%) | (15–30%) | (30–45%) | (45–60%) | (60–80.6%) |
| 14 (13%) | 25 (23%) | 36 (33%) | 28 (26%) | 5 (5%) |
| Population density (inhabitants per km$^2$) | | | | |
| (122–2000) | (2000–4000) | (4000–6000) | (6000–8000) | (8000–15,020) |
| 34 (31%) | 20 (19%) | 31 (29%) | 12 (11%) | 11 (10%) |
| Percentage of green area in the neighborhood | | | | |
| (0.43–20%) | (20–40%) | (40–60%) | (60–80%) | (80–82%) |
| 22 (20%) | 39 (36%) | 25 (23%) | 21 (19%) | 1 (1%) |
| Interquartile coefficient | | | | |
| (65–80) | (80–95) | (95–105) | (105–120) | (120–136) |
| 13 (12%) | 43 (40%) | 27 (25%) | 24 (22%) | 3 (3%) |
| Interquartile asymmetry | | | | |
| (10–20) | (20–25) | (25–30) | (31–36) | |
| 11 (10%) | 26 (24%) | 36 (33%) | 34 (31%) | |
| Net taxable income (€) | | | | |
| (9049–10,000) | (10,000–15,000) | (15,000–20,000) | (20,000–25,000) | (25,000–25,813) |
| 1 (1%) | 8 (7%) | 55 (51%) | 40 (37%) | 4 (4%) |

Table A3 contains the distribution of the multifunctionality (MF) index, which is a composite indicator developed to assess the degree of multifunctionality of the green projects. The MF index combines information on the provision of ecosystem services, recreational opportunities, and social values into a single score. The MF index ranges from 0 to 37, with higher values indicating a higher degree of multifunctionality.

To facilitate interpretation of the results, the MF index scores were categorized into three classes: low (0–11), medium (12–24), and high (25–37) multifunctionality. Table A3 provides the frequency distribution of projects within each of these categories.

**Table A3.** Multifunctionality index results.

| Low MF (0–12) | Medium MF (12–24) | High MF (24–37) |
|---|---|---|
| 19 (18%) | 66 (61%) | 22 (20%) |

Table A4 in Appendix A contains the statistical test results, including one linear regression test between the number of projects per city and the number of inhabitants, as well as four Kruskal–Wallis tests between the multifunctionality (MF) index and the project descriptors.

**Table A4.** Statistical test results.

| Test | Type | *p*-Value | adj R$^2$ |
|---|---|---|---|
| Population city~number of projects | Linear regression | 0.000759 | 0.664 |
| MF index~natural–technical scale | Kruskal–Wallis rank sum test | 0.001058 | / |
| MF index~project size | Kruskal–Wallis rank sum test | $3.735 \times 10^{-5}$ | / |
| MF index~participation | Kruskal–Wallis rank sum test | 0.01542 | / |
| MF index~inclusivity | Kruskal–Wallis rank sum test | 0.0001045 | / |

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
