# Peer review of "Beyond Biodiversity: Eliciting Diverse Values of Urban Green Spaces in Flanders"

_land, doi:10.3390/land12061186_

Round 1

Reviewer 1 Report

The article is interesting and addresses an issue of increasing attention given also the thrust of European and national policies towards solutions based on nature. However, the article is very descriptive and it remains unclear to me what the objective is. The idea is to explore how nature-based solutions are designed and how they relate to social, economic, and environmental elements of the context. In this sense, the article has serious shortcomings: 1. According to what theoretical approach were the codes selected to analyse the projects? Is it a new methodological proposal or is there a reference in this sense in the scientific literature? These questions are even more important for the People category. 2. It would have made much more objective and scientific sense to correlate project codes, and thus their objectives and structure, according to the socio-economic descriptors and the parameters for inclusion described in Tables A2 and A3. 

It is therefore suggested to reprogram the analysis in this direction and on this basis to discuss new findings and new conclusions.

Reviewer 2 Report

159-167: no research hypotheses.

l. 440-445: most of the statistical analyses performed were not significant, therefore, it significantly reduces the scientific value of the publication. Research hypotheses should be put forward and subjected to statistical analysis again.

l. 447-478: the discussion chapter describes the results of the research.

Reviewer 3 Report

This paper addressed the hot topic about relational values associated with urban nature concerning urban green spaces. There are several issues need to be reorganized as follows:

Line 32: Make the exposition tight and clear (right now it is meandering and redundant, particularly in the introduction part of the paper), and make clear what is new, what are the gaps in the literature being addressed, and why doing so is important.

Line 85: Lack of the punctuation.

Line 97: More references (especially in English) are needed to support the statement “Flanders has become Europe's most urbanized region, Flanders has the highest share of urban and built-up areas and the lowest share of natural ecosystems’.

Line 185: Make sure that the contents in table are shown in English for easy reading.

Line 200: If possible, the brief mechanism of “code tree’working can be shown in equations or flow charts.

Line 204: he?

Line 232-233: More details are needed to describe why these six parameters were selected.

Line 238: More details are needed to describe the degree separation for nature gradient.

Line 313: s?

Line 446: If possible, position your research with other contributions to the literature in the discussion section, specifically detailing what is new, different, and/or the same. Try to make discussion much more holistic and simple.

English is fine.

Round 2

Reviewer 2 Report

l. 359-360 - the table should be an illustration of the text 

Author Response

Response to reviewer 3

Dear Reviewer,

Thank you for your feedback and suggestions on the minor revisions. We have addressed the points raised and made the necessary adjustments. We appreciate your valuable input and look forward to your final evaluation.

  1. 359-360 - the table should be an illustration of the text 

We restructured this result to make sure the table is illustrative towards the text. Thank you for pointing out this inconsistency.

Reviewer 3 Report

Line 99: I can not find the corresponding references for the citation [38-40]

Line 232-233: ‘More details are needed to describe why these six parameters were selected.Thanks for pointing this out. We added more information to this explanation.’ It is hard for me to find the relevant added information.

Line 433: ‘Discussion and conclusion’needed to be revised as “Discussion’.  

English Language is fine.
